# High-Fat Diet Impairs Mouse Median Eminence: A Study by Transmission and Scanning Electron Microscopy Coupled with Raman Spectroscopy

**DOI:** 10.3390/ijms22158049

**Published:** 2021-07-28

**Authors:** Ilenia Severi, Marco Fosca, Georgia Colleluori, Federico Marini, Luca Imperatori, Martina Senzacqua, Angelica Di Vincenzo, Giorgio Barbatelli, Fabrizio Fiori, Julietta V. Rau, Antonio Giordano

**Affiliations:** 1Department of Experimental and Clinical Medicine, Marche Polytechnic University, Via Tronto 10/A, 60020 Ancona, Italy; i.severi@univpm.it (I.S.); g.colleluori@pm.univpm.it (G.C.); m.senzacqua@staff.univpm.it (M.S.); angelica-gila@hotmail.it (A.D.V.); g.barbatelli@staff.univpm.it (G.B.); 2Istituto di Struttura della Materia (ISM-CNR), via del Fosso del Cavaliere 100, 00133 Roma, Italy; marco.fosca@artov.ism.cnr.it (M.F.); luca.imperatori@ism.cnr.it (L.I.); 3Department of Chemistry, Università “La Sapienza”, Piazzale Aldo Moro 5, 00185 Roma, Italy; federico.marini@uniroma1.it; 4Department of Odontostomatologic and Specialized Clinical Sciences—Biochemistry, Biology and Physics Section, Via Brecce Bianche, 60131 Ancona, Italy; f.fiori@staff.univpm.it; 5Department of Analytical, Physical and Colloid Chemistry, Institute of Pharmacy, Sechenov First Moscow State Medical University, Trubetskaya 8, Build. 2, 119991 Moscow, Russia

**Keywords:** obesity, energy balance, Arc-ME complex, tanycytes, glial cells, cell junctions, mitochondria lipotoxicity, cyanide, cellular stress

## Abstract

Hypothalamic dysfunction is an initial event following diet-induced obesity, primarily involving areas regulating energy balance such as arcuate nucleus (Arc) and median eminence (ME). To gain insights into the early hypothalamic diet-induced alterations, adult CD1 mice fed a high-fat diet (HFD) for 6 weeks were studied and compared with normo-fed controls. Transmission and scanning electron microscopy and histological staining were employed for morphological studies of the ME, while Raman spectroscopy was applied for the biochemical analysis of the Arc-ME complex. In HFD mice, ME β2-tanycytes, glial cells dedicated to blood-liquor crosstalk, exhibited remarkable ultrastructural anomalies, including altered alignment, reduced junctions, degenerating organelles, and higher content of lipid droplets, lysosomes, and autophagosomes. Degenerating tanycytes also displayed an electron transparent cytoplasm filled with numerous vesicles, and they were surrounded by dilated extracellular spaces extending up to the subependymal layer. Consistently, Raman spectroscopy analysis of the Arc-ME complex revealed higher glycogen, collagen, and lipid bands in HFD mice compared with controls, and there was also a higher band corresponding to the cyanide group in the former compared to the last. Collectively, these data show that ME β_2_-tanycytes exhibit early structural and chemical alterations due to HFD and reveal for the first-time hypothalamic cyanide presence following high dietary lipids consumption, which is a novel aspect with potential implications in the field of obesity.

## 1. Introduction

Obesity, a chronic, non-communicable disease characterized by an excessive body fat accumulation, has spread as an epidemic worldwide [1,2,3]. The combination of overnutrition and sedentary behavior, typical of industrialized countries, progressively leads to the expansion of adipose depots and to fat accumulation in organs not specialized in lipid storage, i.e., liver, skeletal muscle, and pancreas [4,5]. Over time, these alterations induce detrimental mechanical, metabolic, and inflammatory effects that lead to the development of several chronic disorders, including dyslipidemia, insulin resistance, non-alcoholic fatty liver disease, cardiovascular diseases, and some types of cancer [6,7,8].

In mammals, the brain plays a crucial role in energy balance homeostasis and metabolic adaptation to different nutritional and environmental requirements [9,10,11,12]. It receives diet-induced peripheral signals and integrates them with environmental (e.g., visual, taste and olfactory stimuli) and internal (e.g., degree of adiposity, stress and past experience) cues to generate appropriate behavioral, autonomic, and endocrine outputs. Energy metabolism, feeding behavior and body weight are largely controlled by the hypothalamus [9,10,12,13]. Specifically, the hypothalamic arcuate nucleus (Arc), which contains peculiar neuronal populations displaying different neuropeptide expression and functional profiles, is a key player. Here, the orexigenic neuropeptide Y (NPY) and Agouti-related peptide (AGRP)-containing neurons and the anorexigenic pro-opiomelanocortin (POMC)-containing neurons finely coordinate food intake and energy balance and are largely influenced by circulating hormones, nutrients, and metabolites [10,12,13]. Hence, the exploration of all possible routes through which circulating substances gain access to distinctive brain areas involved in energy balance regulation has become a critical field of neuroendocrinology.

A key region in close proximity to the hypothalamic Arc is the median eminence (ME), which is a circumventricular organ lacking the blood–brain barrier (BBB) that is located in the tuberal hypothalamus. Indeed, the ME contains fenestrated capillaries forming the primary plexus of the hypothalamic-adenohypophysial portal system, which allows a rapid delivery of regulatory factors to the adenohypophysis. Furthermore, since the ME lays adjacent to the hypothalamic Arc and forms the base of the third ventricle, it has a strategical anatomical position: it allows the passage of circulating factors to AGRP and POMC neurons, and from fenestrated capillaries into the cerebrospinal fluid, where molecules can reach other hypothalamic and extrahypothalamic areas [10,12,13]. In this region, a crucial function is exerted by tanycytes, radial glia-derived ependymal cells with an elongated shape featuring a single process deepening into the hypothalamic parenchyma [14,15]. Tanycytes are classified into four subtypes (α_1_, α_2_, β_1_, β_2_) based on their location, spatial relationship, and repertoire of junctional complex proteins [15]. β_1_-tanycytes line the infundibular recesses of the third ventricle and send their projections to the pial surface, separating the ME from the Arc. Hence, β_1_-tanycytes are thought to regulate the access of circulating substances extravasated from the fenestrated capillaries of the ME to the Arc neurons [15]. In contrast, β_2_-tanycytes, lining the bottom part of the third ventricle and forming the roof of the ME, are strongly linked by tight junctions that prevent uncontrolled diffusion of substances from the ME milieu to the cerebrospinal fluid [15]. However, they also send their projections on, or in close proximity to, the ME fenestrated capillaries [16]. Such morphological arrangement is believed to subserve the selective and bidirectional transport of molecules between blood and cerebrospinal fluid [14,15]. Importantly, β_2_-tanycytes have been recently suggested to allow the passage of leptin into the cerebrospinal fluid and to be involved in the obesity-induced leptin resistance [16]. However, diet-induced obesity effects on the ME ultrastructural features have been only marginally explored [17]. In rodents, diet-induced obesity develops mostly because of the higher calorie intake occurring during the first 4 weeks of high-fat diet (HFD) [18,19], which is the moment at which body weight segregates from those of controls on chow diet [20]. In addition, hypothalamic inflammation, strictly associated with the excess of dietary lipids, is among the earliest events induced by such dietary protocol and is associated with impaired anorexigenic insulin signaling [18,19,21]. Therefore, the objective of the present study was to assess and characterize the morphological alterations occurring in the ME of mice following 6 weeks of HFD with a particularly high content of lipids (50% of total energy).

The morphological analyses were performed by light microscopy, transmission, and scanning electron microscopy to detect possible cellular and intracellular alterations following HFD-induced obesity. To obtain further insights on and eventually quantify the diet-dependent tissue alterations, we exploited Raman spectroscopy on coronal sections of the Arc-ME complex. Raman spectroscopy is a label free and chemically selective technique that is capable of performing a biochemical map non-invasively of macroscopic specimens at submicrometric resolution. It holds the potential to greatly impact the biomedical field as a powerful clinical investigation tool for diagnosis, providing biochemical information matched with specific morphological features [22,23,24]. In this study, direct spectroscopic evaluation of major Raman features of different groups was performed.

Collectively, our main results show that in the ME of HFD mice, (i) β_2_-tanycytes undergo profound degenerative changes, (ii) the Arc-ME complex exhibits increased lipid and glycogen content, and (iii) the fenestrated capillaries are surrounded by a higher number of collagen fibrils, suggesting that HFD greatly impairs the structure and chemical composition of ME. In addition, the higher cyanide content found in the Arc-ME of HFD-mice in comparison with the control mice reveals a possible additional molecule involved in diet-induced cellular toxicity.

## 2. Results

### 2.1. High-Fat Diet Induces Prominent Alterations in the Median Eminence β_2_-tanycyte Layer

After 6 weeks on an HFD, differences in total body weight were not significant between CTRL and HFD mice, while epidydimal fat depots exhibited a significantly higher weight in HFD than in CTRL (Appendix A).

The ME (squared area in Figure 1A) is characterized by a layered morphological organization that is evident in toluidine blue-stained hypothalamic coronal semithin sections (Figure 1B,C). In CTRL mice (Figure 1B), from the ventricular to the meningeal pial surface, it was possible to distinguish the following: an ependymal layer made up of β_2_-tanycytes (1); an underlying space mostly containing neuronal and glial somata (2); a palisade region occupied by neuronal projections and tanycyte basal processes (3); large, tortuous, and fenestrated capillaries (4); and the meningeal pial covering. A corresponding section of the ME from an HFD mouse (Figure 1C) revealed that even whether the layered organization of the ME was maintained, β_2_-tanycytes were not regularly aligned and displayed disorganized nuclei and a pale cytoplasm, which appeared less stained than in CTRL mice. In addition, the underlying space occupied by neuronal and glial projections and somata appeared deranged in HFD mice compared to the CTRL. The prominent derangement of the β_2_-tanycyte layer found in HFD mice prompted us to assess the number of β_2_-tanycytes by morphometry. A trend for a lower number of β_2_-tanycytes was revealed in the HFD group compared to CTRL, even though the difference was not statistically significant (average number of β_2_-tanycytes in each semithin section: 33.50 ± 3.13 vs. 27.00 ± 1.50, *p* = 0.009, Figure 1D). At higher magnification, the HFD β_2_-tanycyte layer appeared to contain numerous lipid droplets only occasionally present in CTRL condition, and it seemed to contain and/or to be surrounded by empty and larger spaces that were less evident in CTRL condition (compare Figure 1F with Figure 1E).

To better investigate the spatial distribution of β_2_-tanycytes following HFD, we performed a scanning electron microscopy (SEM) analysis on the ME sagittal surface (Figure 1G, analyzed area by SEM). In CTRL mice, the ependymal layer of β_2_-tanycytes appeared regularly organized, with elongated, adjacent cells that shaped a well-defined layer lining the third ventricle (Figure 1H). In contrast, but consistently with what was observed in the semithin sections, the HFD mice displayed less aligned cells with different shapes and sizes and numerous lateral and apical protrusions and vesicles (Figure 1I).

### 2.2. Transmission Electron Microscopy Discloses Numerous Degenerating Aspects in β_2_-tanycytes from HFD Mice

Based on the light microscopy results, the ultrastructural investigation was mainly focused on the β_2_-tanycyte layer. In CTRL mice, tanycytes forming the ependymal layer exhibited elongated and often-indented nuclei as well as an electron-dense and well-differentiated cytoplasm, rich of numerous organelles (Figure 2A,B). The space beneath the ependymal layer contained axons, dendrites, and tanycyte projections and spared neuronal and glial cell bodies (Figure 2A). At their latero-apical surface, β_2_-tanycytes were joined by evident junctions, whose extension was increased by membrane infoldings facing opposing membrane irregularities of the adjacent cell (Figure 2B). In HFD mice, a consistent proportion (about 30% ± 7, n = 3) of β_2_-tanycytes appeared deeply altered. These ependymal cells exhibited roundish nuclei and displayed abnormal enlargements of the extracellular space (Figure 2C). The underlying subependymal layer showed large vacuoles and empty spaces, while the neuropil structures seemed to be weakly packed (Figure 2C). These aspects, namely vacuolization and large extracellular spaces, were suggestive of *subependymal oedema* [25,26], which is a histopathological condition frequently found in hydrocephalus and linked to reduced or degenerated cellular junctions. Interestingly, in HFD mice, tanycytic lateral membranes exhibited reduced infoldings and junctions’ shorter extension (Figure 2D). This qualitative observation was confirmed by morphometric analysis: the length of the portion occupied by tight junctions was significantly lower in HFD mice compared to CTRL (tight junctions’ length in µm in CTRL vs. HFD: 1.86 ± 0.10 vs. 1.20 ± 0.12, *p* < 0.05, Figure 2E). Moreover, morphometric analysis confirmed a higher density of cytoplasmic vesicles in HFD β_2_-tanycytes compared with CTRL (number of vesicles in 100 µm^2^ in CTRL vs. HFD: 22.17 ± 3.73 vs. 37.82 ± 1.87, *p* < 0.05, Figure 2F), whereas the mean size of cytoplasmic vesicles was not significantly affected by HFD (average size in µm^2^ in CTRL vs. HFD: 0.116 ± 0.020 vs. 0.127 ± 0.037, *p* = 0.78, Figure 2G).

At higher magnification, the perikaryon of β_2_-tanycytes of CTRL mice displayed typical features of a well-differentiated cell, rich in free ribosomes, large mitochondria, cisternae of rough endoplasmic reticulum (RER), Golgi complex, and differently sized vesicles (Figure 3A) [15,27,28]. Glycogen was present in a small amount, whereas lysosomes and lipid droplets were only occasionally found. In striking contrast, HFD β_2_-tanycytes were characterized by an electron transparent cytoplasm and by altered/degenerating organelles, numerous vesicles, primary and secondary lysosomes often associated to lipid droplets (Figure 3B). In details, in HFD condition, the mitochondria were very heterogeneous in shape and size, and they presented swellings and internal alterations, including abnormally arranged cristae, absence of cristae in distinctive areas and, sometimes, the presence of myelin-like structures (compare Figure 3D,H with Figure 3C). These mitochondria resembled the swollen or *hydropic* mitochondria that are frequently found when an uncontrolled entry of water (or solutes) occurs in the cell [29]. This cellular event is characterized by progressive changes in mitochondrial morphology: they first enlarge, the cristae disappear, and lastly, the organelle appears empty. Degenerating mitochondria usually undergo mitophagy, which is a strategy adopted to clear the damaged organelle [30]. In HFD β_2_-tanycytes, *hydropic*-like mitochondria were often found in close proximity to lysosomes and autophagosomes, which are both frequently detected in these cells (Figure 3D,H). Accordingly, the density of mitochondria in HFD β_2_-tanycytes was significantly lower compared to CTRL (number of mitochondria in 100 µm^2^ in CTRL vs. HFD: 36.32 ± 2.39 vs. 21.43 ± 1.79, *p* < 0.05, Figure 3I).

β_2_-tanycytes in HFD condition were also characterized by hypertrophic Golgi complex with expansions of cisternae and vesicles (compare Figure 3F with Figure 3E), dilatated and swollen RER cisternae (compare Figure 3H with Figure 3G), and glycogen deposits often attached to organelles. Sporadically, proteinaceous-like material was found into swollen RER cisternae of HFD β_2_-tanycytes (Figure 3H), possibly suggesting intracellular protein degradation processes due to endoplasmic reticulum stress.

### 2.3. Effects of HFD on Other ME Cell Types

Even whether the β_2_-tanycyte layer was the most affected by HFD, also neuronal and/or glial cells located in the subependymal region of the ME occasionally showed signs of degeneration, including an electron transparent cytoplasm and the presence of altered/degenerating mitochondria and numerous vesicles and lysosomes (compare Appendix A with Appendix A). In addition, TEM analysis of the lower ME facing the pia mater revealed numerous collagen fibrils frequently present in the perivascular space around the fenestrated capillaries in HFD condition (compare Figure 4B,D with Figure 4A,C). Finally, cerebrovascular endothelial cells in the fenestrated capillaries of the lower layer of ME in HFD were occasionally characterized by the presence of degenerating aspects i.e., altered mitochondria (Figure 4D), which was an evidence previously described by TEM in BBB-endothelial cells of the ARC in mice exposed to HFD [31].

To better characterize ME alterations, we performed histological staining for collagen and glycogen in ME paraffin sections from CTRL and HFD mice. Sirius Red staining for collagen and PAS staining for glycogen did not show evident differences between CTRL and HFD (data not shown).

### 2.4. HFD Induces Distinctive Biochemical Alterations in the Hypothalamic Arc-ME

To confirm and further characterize the cellular alterations disclosed by the morphological analysis, we exploited Raman spectroscopy to study ME biochemical features. Considering that tanycytes extend their projections into both the median eminence and the Arc, Raman spectroscopy analysis was applied to the Arc-ME complex from CTRL and HFD mice (Figure 5A).

Collectively, a comparison of the averaged Raman spectra from the CTRL and HFD mice revealed evident differences at distinctive excitation wavelengths, with HFD spectra exhibiting bands that were significantly more intense compared to those from CTRL spectra. To establish differences in chemical composition, we referred to references where spectra peaks are associated with specific chemical compounds as shown in Figure 5B.

In particular, in the fingerprint (FP) region, the presence of glycogen within the investigated specimens was supported by the presence of Raman signals at 484, 858, and 1460 cm^−1^. These signals are respectively generated by in plane bending of C-C-C and C-O-C groups in glycosidic bond (484 cm^−1^), C(1)-H(α) bending mode (856/858 cm^−1^), and in plane bending of C-C-C skeletal mode (1460 cm^−1^) [32,33,34,35]. Comparison of these specific optical signatures in HFD and CTRL mean spectra revealed a higher intensity of all the aforementioned bands in the Raman pattern of HFD mice, allowing to infer a higher presence of glycogen in Arc-ME in such dietary conditions (Figure 5C).

Collagen stands for the main component of connective tissues, consisting of a triple helix of entangled polypeptide chains. When observed by Raman spectroscopy, bands attributable to specific amino-acid monomers and normal modes of peptide bond (mainly amide I and amide III) are the most important spectral signature of collagen. Bands attributable to collagen I and III display the amide I bands at about 1661 cm^−1^, while the amide III modes are in the range 1240–1280 cm^−1^. Bands belonging to amino acid are usually observed in the spectrum of collagens. This is the case for the bands of collagen type I identified at 1127, 1105, and 858 cm^−1^ peaks, which are respectively assigned to the δ-NCH, νCC skeletal, and νCC ring modes of proline. The band at 1007 cm^−1^ is attributed to phenylalanine, while the strong band at 1460 cm^−1^ can be attributed to the ν_as_ of COO^−^ moiety of lysine; the shoulder at 1172 cm^−1^ is mainly attributable to C-H modes of tyrosine. The medium band at 747 cm^−1^ corresponds to the δ-COO^−^, ω-COO^−^ mode of phenylalanine [36,37,38]. Interestingly, the comparison between CTRL and HFD mean spectra allowed to highlight a sensible higher intensity of collagen bands in HFD mice, such as the ones at 1460 and 858 cm^−1^ and shoulders at 1172 and 1127 cm^−1^ (Figure 5C).

The intense peak observed at 1748 could be confidently attributed to lipids, phospholipids, and triglycerides (fatty acids) and was more intense in the HFD spectra than CTRL (Figure 5C) [32].

To further characterize lipids composition, spectral features able to identify triacylglycerols and to discriminate saturated and unsaturated fatty acids were assessed. The presence of a band at 1750 cm^−1^ is deemed to stand as one of the most specific signatures of triacylglycerols [39], and a qualitative comparison between CTRL and HFD spectra highlighted a stronger intensity of this band in the HFD group (Figure 5D). The presence of monounsaturated or polyunsaturated fatty acids could be determined by comparing the intensity ratio of Raman bands at shift positions around 1650 and 1450 cm^−1^ [40]. Moreover, the presence of spectral signature at 3000 cm^−1^ could be attributed to =C-H moieties and therefore considered a possible marker for unsaturated fatty acids. A semiquantitative comparison between the ratio of a band’s intensity 1655/1444 cm^−1^ of Raman spectrum of HFD and CTRL mice was performed. The 1655/1444 cm^−1^ intensity ratio returns values of respectively about 0.55 for HFD and about 0.88 for CTRL. A qualitative comparison of high wave numbers (HWN) regions showed a slight but still detectable higher intensity of band at 3000 cm^−1^ in CTRL compared to HFD mice (Figure 5D). Collectively, these analyses showed a relative lower content of unsaturated fatty acids and a relative higher content of saturated fatty acids in HFD mice compared to CTRL (Figure 5D).

Figure 5E shows the HWN region (2100÷3100 cm^−1^). The optical feature at 2248 cm^−1^ represents a peculiar band located in the spectral region usually indicated as the *silent zone* due to the complete lack of Raman signals in biological specimen. This band is attributed to the cyanide group (-CN) [35,41], and it showed an evident higher peak in HFD spectra compared to CTRL (Figure 5E).

Bands at 2850 and 2878 cm^−1^, attributed to the symmetric and antisymmetric stretching vibrations C–H of the methylene group CH_2_ of proteins [35,39], did not differ between groups. The band at 2930 cm^−1^, attributed to the antisymmetric stretching C–H vibrations of aromatic groups which are typical of glucoside biomolecules, did not differ between groups [34,35,39,41] (Figure 5E).

In order to confirm the effect of the different diets on the Raman signals measured on the Arc-ME complex, a PLS-DA discriminant classification model was built from the two classes involved after spectral normalization to unit area and mean centering (Figure 5F). The optimal complexity of the model, evaluated by a 10-fold cross-validation, was found to be seven latent variables. This optimal model was able to correctly classify 79.5% of the spectra from the CTRL and 91.9% of the profiles recorded on the HFD mice. When applied to the validation spectra, the model provided 90.9% and 90.0% prediction accuracy for the CTRL and the HFD spectra, respectively.

Interpretation of the models in term of the spectral regions mostly responsible for the differences can be carried out by inspecting the values of the VIP coefficients and those of the regression coefficients. Indeed, on one hand, variables having a VIP score higher than one are usually considered as contributing significantly to the definition of the PLS-DA model and therefore to the observed discrimination. On the other hand, by checking the values (and the sign) of the regression coefficients, one could postulate whether a particular variable is on average higher in one class or another. In the present case, variables higher in the HFD correspond to a positive regression coefficient, while the negative ones are associated to predictors having a higher intensity for CTRL. The regions of the Raman spectra considered to be significant according by the VIP scores and the associated values of the regression coefficients are displayed in Figure 5F and superimposed to the mean spectra of the training samples. The colored areas under the mean spectroscopic signal indicate the bands that contribute significantly to the classification model; i.e., the bands can be considered as more discriminant between the HFD and the control individuals. On the other hand, the color of the areas below the curve identifies the sign of the regression coefficient, following the interpretation reported above: red indicates a positive regression coefficient (i.e., intensity which is, on average, higher for the HFD), and blue is associated with negative values (i.e., intensity which is, on average, higher for the CTRL).

## 3. Discussion

The ME has been recognized as a critical area involved in the regulation of the energy balance [16]. In particular, ME tanycytes can sense peripheral molecules, allow their exchange from blood to key regions modulating energy homeostasis (e.g., Arc) and, as a consequence, govern the adaptive acute response to distinct nutritional challenges [16,42]. In this study, we demonstrated that 6 weeks of HFD led to significant alterations of ME tanycytes consisting of spatial derangement, organelles anomalies, and higher content of lipids compared to normo-fed controls. Such alterations, revealed by the combination of electron microscopy and spectroscopy techniques, were evident in β_2_-tanycytes. Given β_2_-tanycytes strict contact with fenestrated capillaries, and their role as glucolipid sensors, it is possible that such highly differentiated cells are damaged earlier and more than other cytotypes within the same region as a consequence of the HFD-induced elevation in circulating lipid and glucose. The resulting tanycytic stress and dysfunction may ultimately lead to cell death, contributing in some extents to hypothalamic inflammation, which is a phenomena that may also explain the documented proliferative events observed in this region in HFD conditions [16]. The association between HFD and hypothalamic gliosis is in fact widely recognized [19] and was demonstrated to parallel alterations in lipid metabolism [42]. Accordingly, in our study, higher lipids content was visible within β_2_-tanycytes, while greater lipids, glycogen, and collagen amounts were detected by Raman spectroscopy in the Arc-ME area. In particular, higher intensity of Raman bands corresponding to triglycerides and saturated fatty acids were observed in HFD compared to normo-fed controls who instead displayed higher relative content of unsaturated fatty acids. Based on our observations by TEM, we mainly attributed the elevated lipid quantity detected in the Arc-ME area to ME-tanycytes as opposed to Arc. Such speculation is supported by the results of Hofmann et al., who documented that lipids localize mainly in tanycytes and ependymal layers in normal condition and increase within the same cell types as a consequence of HFD [42]. Importantly, differently from the unsaturated forms, saturated fatty acids were reported to induce mitochondrial impairments and altered expression of cytoskeleton regulators in N42 hypothalamic neurons in vitro [43]. Interestingly, the authors of the same study reported altered hypothalamic expression of mediators of stress response, glucose metabolism, cytoskeleton, and synaptic plasticity after 3 days of HFD in vivo [43]. The impaired glucolipid profile induced by obesity is associated with both inflammation and cellular toxicity [4,5,8], with some tissues being affected earlier than others. Hypothalamic inflammation and dysfunction were demonstrated to be initial events following diet-induced obesity [18,19], occurring even before peripheral inflammation and substantial weight gain [19]. Specifically, the hypothalamic Arc-ME was demonstrated to be the area mainly involved in such a process, which is a phenomenon reported in rodents and also documented in human medio-basal hypothalamus [19]. Differently from other circumventricular organs, the ME rapidly responded to the presence of a high amount of dietary fatty acids, with a significant increase in inflammatory transcripts after 1 and 4 weeks of HFD [17]. Importantly, such inflammatory status was paralleled by an altered expression of regulators of BBB integrity and higher BBB permeability, which is a finding coherent with the widely documented hypothalamic dysfunction following such dietary intervention [17,18,19]. HFD-induced ME alterations display a biphasic trend: early signs of cellular stress are in fact followed by transient improvements (neuroprotective response to limit the damage) and then again by evidence of injury upon sustained insult [17,19]. To avoid the nadir of such biphasic trend, we studied the effect of 6 weeks of HFD and demonstrated that it results in ME–ultrastructural alterations and profound derangements of β_2_-tanycytes i.e., shorter cellular junctions, altered membrane foldings, high prevalence of large, empty vesicles, and electron-transparent intra- and extracellular spaces, signs of cellular stress such as high lysosomes and autophagosomes contents, RER and Golgi apparatus enlargements, and mitochondrial swellings and derangement. Remarkably, mitochondrial density was lower in β_2_-tanycytes of HFD mice who, differently from controls, also displayed the presence of larger Raman bands corresponding to the cyanide group (-CN). Cyanide is recognized as an environmental toxin primarily hampering mitochondria respiration [44,45]. Nevertheless, in its protonated form (HCN), cyanide is also generated by specific cell types within the body and was proposed as a gasotransmitter (similarly to NO, CO, and H_2_S), even though its physiologic role has not been fully characterized yet [44,46,47,48]. Cyanide was reported to be produced by mitochondria’s neurons and modulate the response to different stimuli, possibly involving neurotransmitters release, brain metabolism, and oxidative processes [44,46,48]. Neuronal HCN release was in fact reported to occur in response to hydrogen peroxide or glycine administration [44]. Interestingly, rats hypothalamus was among the brain regions having the highest cyanide production in response to opiate agonists [49]. Hence, it is possible that, at first, a low amount of cyanide is produced in mice brain to respond to the HFD-induced metabolic challenge and secondly, upon sustained production, cyanide excess contributes to Arc-ME injury. Such interpretation is in line with a recent study demonstrating that low cyanide amounts promotes HepG2 mitochondrial respiration, oxygen consumption, and cells proliferation, while high levels result in mitochondrial stress and cellular toxicity [47]. Given the potential cyanide role as a gasotransmitter [47], the diet-induced increase in brain cyanide and the potential biochemical implications deserve further investigation.

In our study, a higher number of empty intracellular vesicles and wider electron-transparent cytoplasmic areas, suggestive of the broad presence of intracellular water, were evident in β_2_-tanycytes of HFD mice compared to normo-fed controls. Extracellular water excess was also evident between β_2_-tanycytes and within the subependymal layer and displayed similar aspects as the ones observed in the hydrocephalus [25,26,50]. Among the roles of β_2_-tanycytes, there is the modulation of solutes passage from the liquor to blood and vice versa [15]. Thus, it is possible that the above-mentioned cellular stress resulted in impairments in organelles and junction’s organization, and that such anomalies deeply compromised tanycytes function, leading to an altered ability to guarantee molecule exchange and osmotic balance. Therefore, the detected alterations in tanycytes’ shape, size, and orientation could be attributed not only to cellular stress per se but also to the excess of intracellular water, which are all phenomena likely leading to cell death.

Higher collagen content following HFD was revealed by Raman spectroscopy and detected by TEM around blood vessels. On the other side, although histochemical staining confirmed the localization of collagen around the fenestrated capillaries of the ME, it did not show an evident higher level of staining in HFD compared to CTRL, which was possibly due to the low sensitivity of the technique. However, hypothalamic hypervascularization and micro-angioarchitecture anomalies consisting of increased extracellular matrix protein deposition, collagen IV, basal membrane size and vascular impairments were recently documented [51] and are consistent with other report [31]. Interestingly, diet-induced hyperleptinemia was recently demonstrated to lead to alteration in the HIF1-α/VEGF signaling in hypothalamic astrocytes, which is in turn responsible for the obesity-induced hypertension [51]. Thus, it is possible that the detected Arc-ME increase in collagen is a consequence of the obesity-induced hypothalamic microangiopathy, which is a phenomenon with critical pathophysiologic implications.

Signs of stress in some cells located in ME lower layers were also visible after 6 weeks of HFD. Given their glucosensory capacity and access to circulating metabolites and hormones, tanycytes have been proposed to modulate the activity of hypothalamic neurons [43]. Tanycytes selective ablation within the ME were in fact associated with increased adiposity [52]. On the other side, HFD-induced hypothalamic dysfunctions were demonstrated to cause anomalies in the regulation of energy intake and expenditure, exacerbating the increase in adiposity induced by diet [16]. According to our study, ME β_2_-tanycytes are relevant and early targets in the context of HFD-induced toxicity, which is a phenomenon possibly underlying the alterations in hypothalamic energy homeostasis regulation following diet-induced obesity [4].

## 4. Conclusions

In conclusion, even though this study cannot provide mechanistic insight into the detailed ME anomalies, it strongly suggests that ME β_2_-tanycytes are early targets of HFD-induced toxicity and that their degeneration may represent the *primum movens* of hypothalamic dysfunction in obesity. This is the first study documenting a detailed ultrastructural characterization of HFD-induced tanycytes alterations and reporting the Arc-ME biochemical variations following such intervention. Furthermore, although a cyanide source could not be identified with the employed techniques, this is the first report evidencing increased cyanide content in the Arc-ME of HFD-fed mice, which is a novel and unexplored aspect that holds critical implications in the field of obesity. Additional studies providing mechanistic insights are needed to confirm and go in depth into our findings.

## 5. Materials and Methods

### 5.1. Animals, Experimental Conditions, and Tissue Processing

Four-week-old CD1 Swiss male mice were purchased from Charles River (Lecco, Italy) and housed individually in plastic cages under constant environmental conditions (12 h light/dark cycle at 22 °C) with ad libitum access to food and water. After one week of acclimatization, animals were weighed and divided into two groups with approximately similar mean body weight: one group (n = 6; CTRL) was fed with a standard diet (Charles River; 19 kJ% from fat, 50 kJ% from carbohydrates and 31 kJ% from proteins), while the remaining mice (n = 9; HFD) was fed with a high-fat diet (Charles River; 50 kJ% from fat, 30 kJ% from carbohydrates and 20 kJ% from proteins). These feeding conditions were maintained for 7 weeks, and individual weight was measured weekly.

After the dietetic treatment, mice were anesthetized with 300 mg/kg 2,2,2-tribromoethanol and perfused transcardially with 4% paraformaldehyde in 0.1 M phosphate buffer (PB), pH 7.4. Brains were carefully removed from the skull, post-fixed with the same fixative solution for 24h at 4 °C, and washed in PB. For TEM application, free-floating coronal sections (200 µm thick) were obtained with a Leica VT1200S vibratome (Leica Microsystems, Vienna, Austria) and kept in phosphate-buffered saline (PBS), pH 7.4, at 4 °C until use. For SEM application, 2 mm thick coronal brain sections including the ME were obtained with the use of a brain matrix (ASI Instruments, Warren, MI, USA) and further processed as described below. For histological staining, brains were embedded in paraffin.

### 5.2. Transmission Electron Microscopy

Free-floating 200 µm thick paraformaldehyde-fixed coronal brain sections (Bregma from −1.46 to 2.18 mm, according to Paxinos and Franklin, 2001) were incubated in 2.5% glutaraldehyde (120 min at 4 °C), thoroughly washed in PB, and post-fixed in 1% OsO_4_ (60 min at 4 °C). After dehydration in ethanol and infiltration in Epon–Spurr resin, sections were flat-embedded between two Aclar (Sigma-Aldrich, 8F119)-coated coverslips. After resin polymerization, small tissue blocks (about 2 mm in size) containing the median eminence were selected by light microscopy, sliced with a lancet, glued to blank epoxy resin, and sectioned with an MTX ultramicrotome (RMC, Tucson, AZ, USA). Semithin sections (1 µm thick) were stained with toluidine blue. Thin sections (70–80 nm thick) were stained with lead citrate and examined in a Transmission Electron Microscope (CM10, Philips, Eindhoven, The Netherlands).

### 5.3. Scanning Electron Microscopy

Coronal brain sections (2 mm thick) were fixed in 2% glutaraldehyde–paraformaldehyde overnight at 4 °C and then sliced under a surgical microscope to isolate the region of interest to be observed by SEM. Then, samples were post-fixed in 1% OsO_4_ in phosphate buffer (PBS) for 60 min at 4 °C, washed in PB, and dehydrated with increasing ethanol concentrations. Samples exsiccation was performed following the *critical point drying*, consisting of the incubation in increasing HMDS:ethanol concentrations, until embedding in 100% HMDS. Samples were mounted on aluminum stabs using cell adhesive carbon disks and underwent a sputtering coating procedure with gold before proceeding with the SEM observation by Tescan Vega3 LMU.

### 5.4. Sirius Red and PAS Staining

Picro Sirius red staining was performed to study the ME collagen presence and distribution. In brief, paraffin-embedded sections (3–4 µm) were first dewaxed and hydrated; then, they were incubated in Sirius Red 0.1% in Sat’d Picric Acid for 60 min at room temperature, rinsed in water, contrasted with hematoxylin for 2 min, rinsed again in water, dehydrated, cleared, and finally mounted with Eukitt.

PAS staining was exploited to study ME glycogen content and distribution. Briefly, paraffin-embedded sections (3–4 µm) were first dewaxed and hydrated; then, they were incubated in 0.5% Periodic Acid Solution for 5 min. After rinsing in water, sections were placed in Schiff Reagent for 15 min, rinsed again in water, counterstained with hematoxylin for 2 min, washed in tap water, dehydrated, cleared, and finally mounted with Eukitt.

### 5.5. Morphometric Analysis

Morphometric data were obtained from three CTRL mice and three HFD mice. For each animal, two 200 µm thick coronal ME semithin sections were analyzed, and for each section, two semithin sections and three ultrathin sections were observed. Toluidine blue-stained semithin sections were used to assess the number of nuclei of β_2_-tanycytes in the ependymal layer. On the other side, ultrathin sections acquired in a Philips CM10 Transmission Electron Microscope, always examined at a magnification ranging from 25,000× to 19,000×, were used to study the cytoplasmic mitochondrial density (number of mitochondria/100 µm^2^ of cytoplasmic area), vesicles density (number of vesicles/100 µm^2^ of cytoplasmic area), size of cytoplasmic vesicles (µm^2^), and the average tight junction length (µm). See Peruzzo et al., 2004 [28] for a tight junction description. About 30 β_2_-tanycytes for each condition were analyzed. Areas were calculated using ImageJ software version 1.52a (NIH, Bethesda, MD, USA). All images from CTRL and HFD mice were processed in parallel and morphometric evaluations on acquired microscopical fields were performed in a blind manner.

All values in graphs are expressed as mean ± standard error of the mean. Data were analyzed for significance with GraphPad Prism (version 8) using an unpaired Student’s *t*-test. The threshold for significance was set at *p* < 0.05.

### 5.6. Raman Spectroscopy Measurements

Raman spectra were recorded from 200 µm thick paraformaldehyde-fixed coronal brain sections put on a pre-cleaned glass by means of the DXR Raman microscope (Thermo Fisher Scientific, Wallthman, MA, USA) under the following conditions: 532 nm laser source; 200–3400 cm^−1^ full range grating; 10× and 50× objectives; 25 μm confocal pinhole; 5 (Full Width Half Maximum) cm^−1^ spectral resolution. No pre-treatment was done on samples. As a first step, collection of low-magnification (10×) mosaic images was carried out to get generic information on tissue morphology. This procedure allowed us to identify the regions of interest (ROIs). Afterward, the ROI containing the ME was selected and investigated at high magnification (50×), recording Raman patterns. All spectra presented in this work were collected with the 50× objective. A laser power of 8 mW, measured at the sample, was applied as a suitable compromise between the signal quality and undesired tissue burning. The exposure time was 1.0 s. At least 30 exposures were averaged to obtain a better Signal-to-Noise Ratio (SNR). A 5th-order polynomial correction via the modified Polyfit software package was used to compensate tissue fluorescence. Laser spot size was about 700 nm (50× objective). Tissue areas from approximately 0.5 × 1.5 mm^2^ were investigated. The applied step size was about 50 and 25 μm (for less homogeneous zones). Multiple measurements were performed at different regions of the same tissue.

A total of 8 maps were recorded, and about 1000 spectra were collected for each map (Appendix A). For each map, the average spectrum was accomplished using a selection of patterns collected upon the ROI. Asymmetric least square method was used for baseline correction to remove further residual fluorescence contributions. One average and one SD spectra for the CTRL and HFD groups were obtained from further averaging the above-mentioned 8 spectra. 500÷1800 cm^−1^ FP and 2100÷3100 cm^−1^, HWN and part of silent zone regions of spectra were considered, as containing useful information. The gray area represents standard deviation values for each averaged spectrum.

### 5.7. Statistical Analysis of Raman Spectroscopy Data

Raman measurements were performed on 3 CTRL mice and 4 HFD mice. For each animal, two 200 µm thick coronal sections of the Arc-ME were analyzed. First, spectra within each image were pretreated by penalized asymmetric least squares correction (PLS), to remove further residual fluorescence contributions, to correct the effects of a non-constant baseline and, at the same time, smoothing the signal. Then, ROIs corresponding to the Arc-ME were manually extracted from each image, to build the dataset to be considered for the subsequent chemometric analysis. Indeed, all the ROIs were joined into a single data matrix having dimensions of 7409 (number of spectra) × 3180 (number of wavenumbers). This dataset was further split in two subsets, the training, and the test sets, to be used for model building and validation, respectively. In order to guarantee that both subsets were equally representative, the duplex algorithm (DA) was individually applied to the data of each class, leading to a training set of 5600 spectra and a test set with the remaining 1809 spectra.

In order to identify and validate spectral regions that could differentiate the HFD obese from the CTRL samples, a classification approach based on the partial least squares discriminant analysis (PLS-DA) algorithm was adopted [53]. PLS-DA is a classification method based on the PLS regression technique [54], and it was devised to allow calculating models also for those situations in which more traditional classification approaches could suffer from ill-conditioning (e.g., when there are more variables than samples, or when variables are highly correlated, as in the case of spectroscopic data). Indeed, PLS regression overcomes these limitations by projecting the predictors onto a low-dimensional space of orthogonal latent variables (LVs), which have the characteristic of being the direction of maximum covariance with the responses to be predicted. To be able to use a regression technique (PLS) to achieve classification (PLS-DA), it is necessary to introduce a dummy response y, which codes for class belonging. In the case of problems involving two categories, this is accomplished by coding the first class as y = 1 and the second as y = 0; then, any new sample is classified by setting a threshold, which is usually 0.5. If the predicted y is higher than the threshold, the sample is classified as class 1; otherwise, it is classified as class 2.

## Figures and Tables

**Figure 1 ijms-22-08049-f001:**
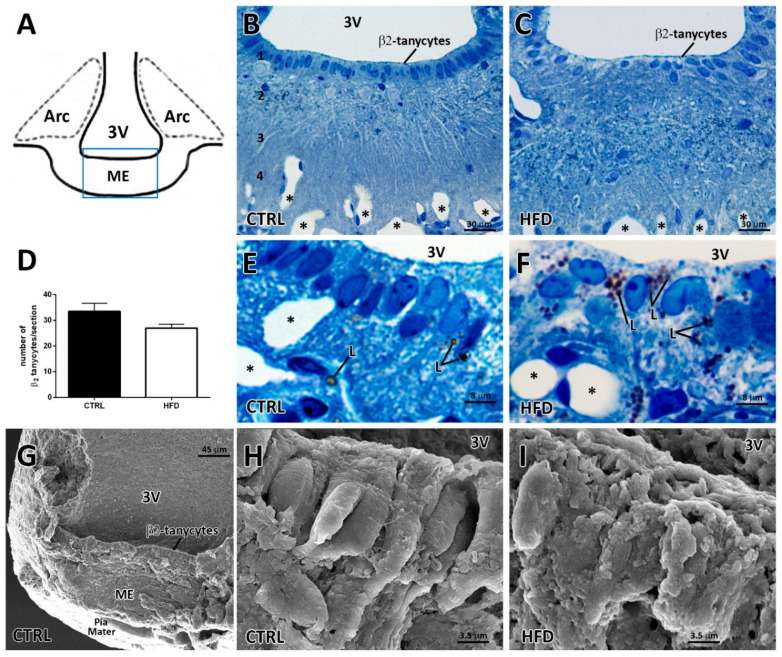
Morphological aspects of the median eminence from normo-fed controls (CTRL) and high-fat diet (HFD) mice. (**A**): Schematic picture of a coronal section of mouse hypothalamus, showing the studied area (squared). (**B**,**C**): Light microscopy of toluidine blue-stained hypothalamic coronal semithin sections from a CTRL (**B**) and a HFD (**C**) mouse. The CTRL median eminence (ME), from the ventricular to the pial surface, shows a layered organization: an ependymal layer of β_2_-tanycytes (1); an underlying space mostly containing neuronal and glial somata (2); a palisade region of tanycytes processes (3); fenestrated capillaries (4) (capillaries: asterisks). In the HFD mouse (**C**), the layered organization is still present, but β_2_-tanycytes display disorganized nuclei and less-stained cytoplasm. (**D**): Number of β_2_-tanycytes/section in CTRL and HFD mice. Data (n = 3 mice for each condition) are reported as mean ± standard error (*p* < 0.05 by unpaired student’s *t*-test). (**E**,**F**): High magnification of the β_2_-tanycyte layer in semithin sections from CTRL and HFD mice, respectively. L, lipid droplets; *, capillaries. (**G**): Scanning electron microscopy image of the ME at low magnification from a CTRL mouse. (**H**,**I**): Scanning electron microscopy images of β_2_-tanycytes in a CTRL and a HFD mouse, respectively. 3V, third ventricle; Arc, arcuate nucleus.

**Figure 2 ijms-22-08049-f002:**
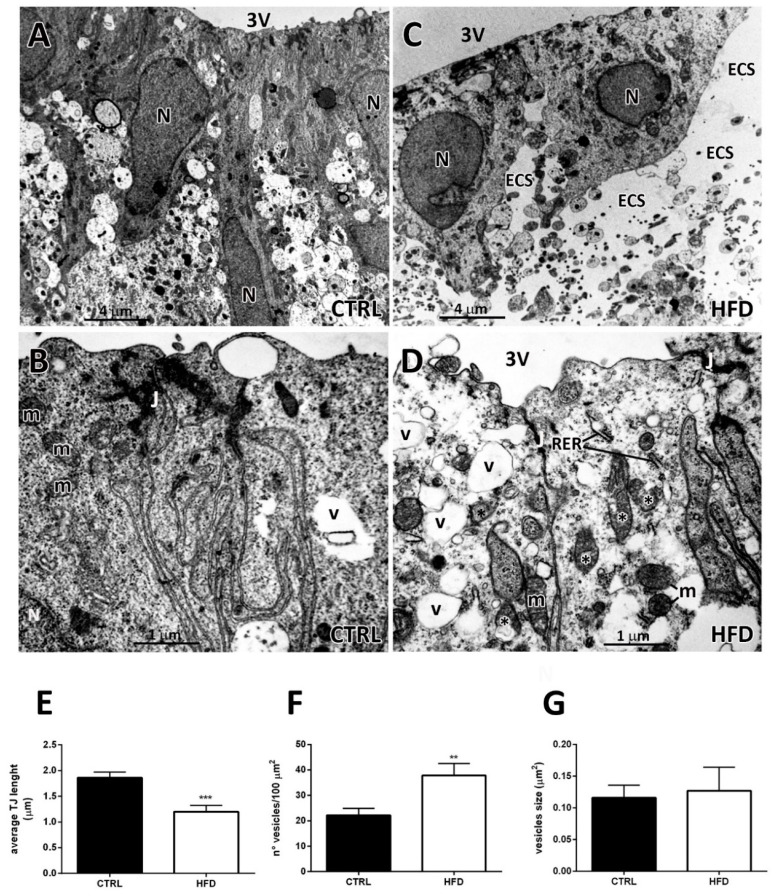
Transmission electron microscopy and morphometric analyses of β_2_-tanycytes from normo-fed controls (CTRL) and high-fat diet (HFD) mice. (**A**): β_2_-tanycytes from CTRL mice display elongated and regular nuclei (N) and a dense, organelle-rich cytoplasm. The underlying space contains numerous axons, dendrites, and tanycyte projections. (**B**): At higher magnification, two adjacent β_2_-tanycytes from a CTRL mouse show electron dense and well-differentiated cytoplasm, containing normal mitochondria (m), some vesicles (v), and well-developed junctions (J) linking adjacent cells. Strongly electrodense tight junction portions and long and convoluted infoldings are present. (**C**): β_2_-tanycytes from HFD mice display cuboidal nuclei and electron transparent cytoplasm. The underlying space (ECS, extracellular space) appears enlarged, empty (electron-transparent), and the neuropil structures are weakly packed. (**D**): At higher magnification, two adjacent β_2_-tanycytes from an HFD mouse are linked by short electrodense thigh junctions (J), lack infoldings and display cytoplasmic alterations. In these cells normal mitochondria (m) are present together with degenerating ones (*). In addition, numerous, large vesicles (v) and often dilated rough endoplasmic reticulum (RER) cisternae are present. (**E**): Average tight junction length in β_2_-tanycytes from CTRL and HFD mice. (**F**): Average number of cytoplasmic vesicles in β_2_-tanycytes from CTRL and HFD mice. (**G**): Size of cytoplasmic vesicles in β_2_-tanycytes from CTRL and HFD mice. Data in (**E**–**G**) (n = 3 mice for each condition) are mean ± standard error, ** *p* < 0.01, *** *p* < 0.001 compared to control mice (unpaired Student’s *t*-test).

**Figure 3 ijms-22-08049-f003:**
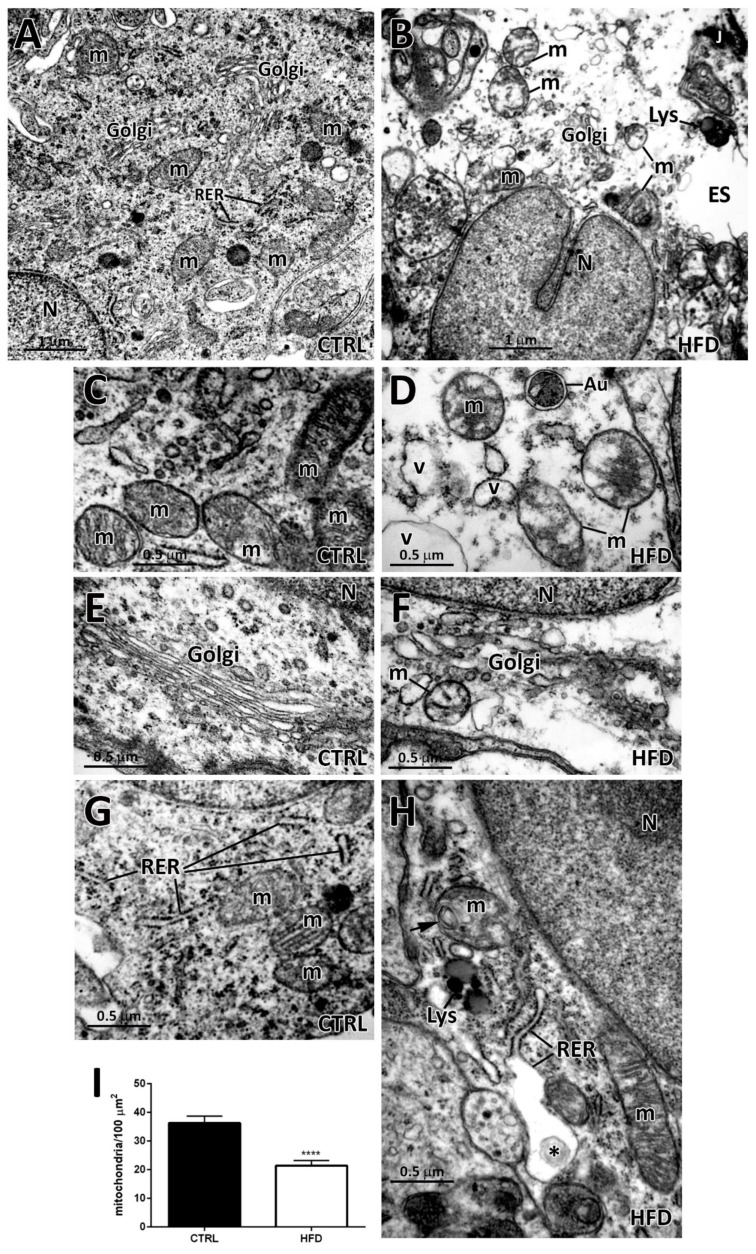
Intracellular ultrastructural features of β_2_-tanycytes from normo-fed controls (CTRL) and high-fat diet (HFD) mice. (**A**): Representative image of the cytoplasm of a β_2_-tanycyte from a CTRL mouse: normal mitochondria (m), Golgi complexes (Golgi), rough endoplasmic reticulum (RER), and few vesicles. (**B**): Representative image of a degenerating β_2_-tanycyte cytoplasm from an HFD mouse: altered mitochondria (m), Golgi complex (Golgi), and presence of lysosomes (Lys), and numerous vesicles. In the right part of the figure, the extracellular space (ES) is abnormally enlarged (J, tight junction). (**C**,**D**): Comparison between normal mitochondria (m) in a CTRL β_2_-tanycyte (**C**) and swollen mitochondria (m) with few cristae and pale internal matrix in a HFD β_2_-tanycyte (**D**). Frequently, autophagosomes (Au) and vesicles (v) of heterogeneous shape and size are present near to degenerating organelles. (**E**,**F**): Comparison between a normal Golgi complex adjacent to the nucleus from a CTRL β_2_-tanycyte (**E**) and a dilated Golgi complex adjacent to the nucleus from a HFD β_2_-tanycyte (**F**). (**G**,**H**): Comparison between normal rough endoplasmic reticulum (RER) in a CTRL β_2_-tanycyte (**G**) and a dilated and swollen RER in a HFD β_2_-tanycyte (**H**). Note the presence of proteinaceous-like material in the swollen RER (asterisk) and of a myelin-like figure (arrow) in a mitochondrion of the HFD β_2_-tanycyte. (**I**): The density of mitochondria in β_2_-tanycytes from CTRL and HFD mice is shown. Data (n = 3 mice for each condition) are presented as mean ± standard error; **** *p* < 0.0001 by unpaired Student’s *t*-test. N, nucleus.

**Figure 4 ijms-22-08049-f004:**
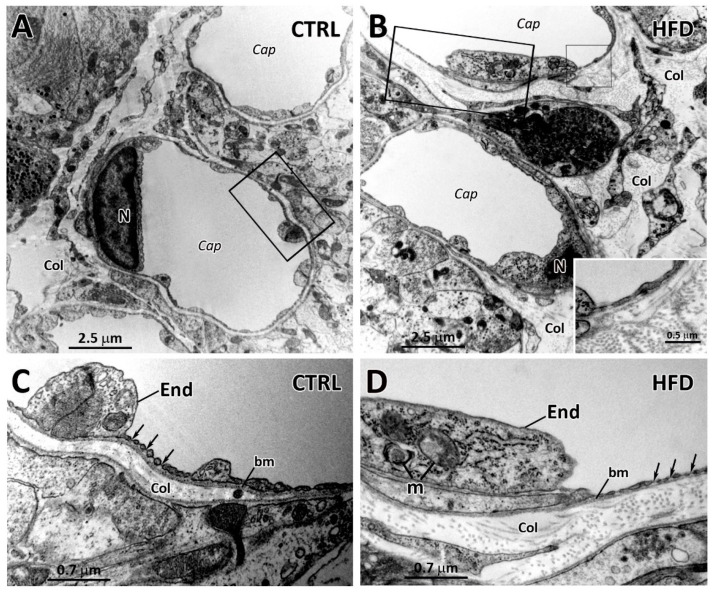
Transmission electron microscopy of capillaries located in the external portion of the median eminence from normo-fed controls (CTRL) and high-fat diet (HFD) mice. (**A**,**B**): Low magnification of median eminence capillaries (Cap) from a CTRL (**A**) and a HFD mouse (**B**). In (**A**), few collagen fibrils (Col) are visible in the perivascular space, whereas in (**B**) higher presence of perivascular collagen fibrils (Col) is visible. The inset of (**B**) is the enlargement of the smaller area framed in (**B**) and shows collagen fibrils at higher magnification. (**C**,**D**): Higher magnifications of the areas framed in (**A**,**B**), respectively, showing fenestrated walls of the capillaries (arrows) and collagen fibrils (Col) distributed in the perivascular spaces, below the basal membranes (bm) of endothelial cells (End). In (**D**), degenerating mitochondria (m) in the cytoplasm of an endothelial cell are also visible. N, nucleus.

**Figure 5 ijms-22-08049-f005:**
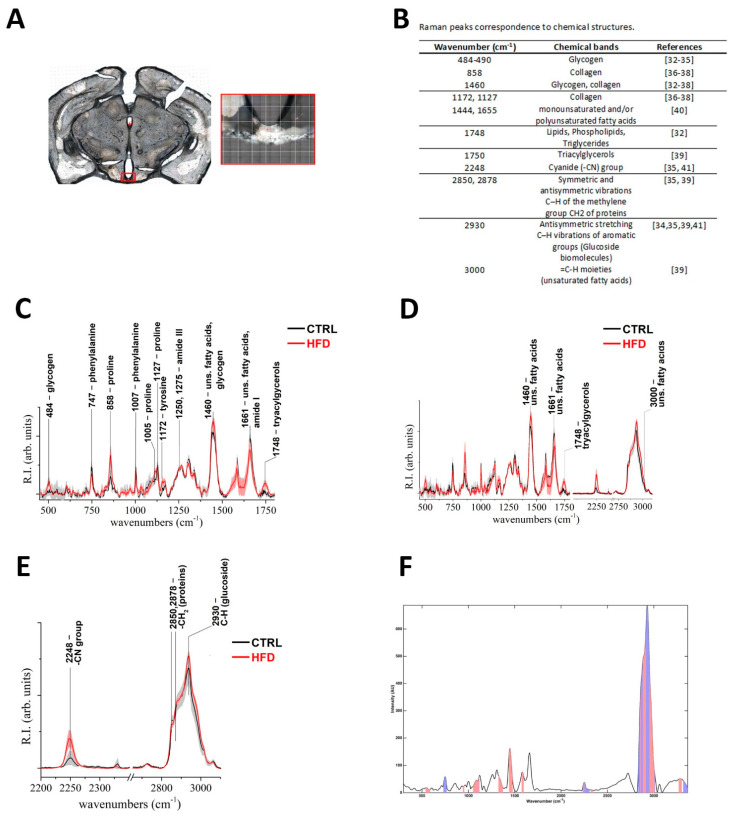
Raman spectroscopy analysis of Arc-ME complex from normo-fed controls (CTRL) and high-fat diet (HFD) mice. (**A**): Representative coronal section showing the analyzed region of interest, i.e., the Arc-ME complex, framed by red lines. (**B**): Raman peaks correspondence to chemical structures. (**C**): Fingerprint (FP) (400÷1800 cm^−1^) region of averaged Raman spectra collected from CTRL (black line) and HFD (red line) mice. (**D**): FP and high wave numbers (HWN) regions of averaged Raman spectra from CTRL (black line) and HFD (red line) mice are reported with more specific positions labeled for chemical attributions for lipids, fatty acids, and triacylglycerols. (**E**): HWN and part of the “silent zone” (2100 ÷ 3100 cm^−1^) regions of averaged Raman spectra from CTRL (black line) and HFD (red line) mice. In C, D, and E, gray and light-red areas represents the standard deviation values for each averaged spectra from CTRL and HFD mice, respectively. The shift positions of the most intense bands are labeled. (**F**): Partial least-squares discriminant analysis (PLS-DA) model of HFD vs. CTRL. Identification of the regions significantly contributing to the discrimination among the classes according to the values of the associated VIP indices (colored areas under the mean spectrum of the training samples); the colors, red and blue, indicate the sign of the corresponding regression coefficients, positive and negative, respectively.

## Data Availability

Not applicable.

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
