# Peer review of "High-Fat Diet Impairs Mouse Median Eminence: A Study by Transmission and Scanning Electron Microscopy Coupled with Raman Spectroscopy"

_ijms, 2021, doi:10.3390/ijms22158049_

Round 1
Reviewer 1 Report
The Authors of the paper have tried to gain insights into the early hypothalamic diet-induced alterations. Adult CD1 mice fed a high-fat diet for six weeks were studied and compared with normo-fed controls. Transmission and scanning electron microscopy and histological staining were employed for morphological studies of the ME, while the Raman spectroscopy was applied for the biochemical analysis of the Arc-ME complex. This manuscript is a significant contribution to the scientific discussion about effects of high-fat diet and regulating energy balance in mammals. It has high scientific quality. Text editing and minor language revisions should be made.
Author Response
REVIEWER 1: The Authors of the paper have tried to gain insights into the early hypothalamic diet-induced alterations. Adult CD1 mice fed a high-fat diet for six weeks were studied and compared with normo-fed controls. Transmission and scanning electron microscopy and histological staining were employed for morphological studies of the ME, while the Raman spectroscopy was applied for the biochemical analysis of the Arc-ME complex. This manuscript is a significant contribution to the scientific discussion about effects of high-fat diet and regulating energy balance in mammals. It has high scientific quality. Text editing and minor language revisions should be made.
A: We deeply thank the reviewer for the positive comments. According to his/her suggestion, we thoroughly read the manuscript and made some language revisions.
Reviewer 2 Report
This article presents a nice experimental description of the change that occurred in the median eminence due to a high-fat diet in a CD1 mouse model. However, I find, it will be difficult for the readers to understand the scientific message of this article. So, I would recommend this article a major revision before publication.
Major reason:
(1) Figures 1, 2, 3, 4 and supplementary figure 2 are in black and white and the writings in the figure are also in black and white. The authors should come back with a better representation of the figures at least with white background for the writing part.
(2) It seems to me, Figure 4; C, D, and E plot the same results more than once and Figure 4, F is very hard to understand. It would be a much better figure 4 if the peaks are labelled with the proper analytes (name) of interest.
(3) I do not find a separate conclusion section in the article.
Minor comment:
There are some unnecessary "." at the section headings of introduction, results, and discussion.
Author Response
REVIEWER 2: This article presents a nice experimental description of the change that occurred in the median eminence due to a high-fat diet in a CD1 mouse model. However, I find, it will be difficult for the readers to understand the scientific message of this article. So, I would recommend this article a major revision before publication.
Major reason:
(1) Figures 1, 2, 3, 4 and supplementary figure 2 are in black and white and the writings in the figure are also in black and white. The authors should come back with a better representation of the figures at least with white background for the writing part.
We thank the reviewer for his/her comment. Following his/her suggestion, the electron microscopy figures 1,2,3,4 and Suppl. figure 2 have been changed in the revised version of the manuscript.
(2) It seems to me, Figure 4; C, D, and E plot the same results more than once and Figure 4, F is very hard to understand. It would be a much better figure 4 if the peaks are labelled with the proper analytes (name) of interest.
We thank the reviewer for his/her comment. We agree with the reviewer that panels C, D and E of Figure 5 report the same spectral Raman regions. However, while the panel C provides a general overview of the most intense detected Raman peaks, panels D and E detail specific ranges from the whole spectral region (500-3100 cm-1). We would prefer such graphical presentation to allow the reader to better focus on the two major spectral features differentiating the two experimental conditions. Anyway, we leave to the editor the final decision.
As suggested by the Reviewer, the graphical aspect of figure 5F has been modified. The regions identified by VIP>1 have now been highlighted with different colors associated to the sign of the regression coefficient. Figure caption of figure 5F has been modified accordingly and a detailed description of the figure has also been added to the text.
Finally, in the revised version of our manuscript Figures 5 C, D and E have been changed and analytes of interest have been added near the corresponding peaks.
(3) I do not find a separate conclusion section in the article.
We have added a conclusion section in the revised version of the manuscript.
Minor comment:
There are some unnecessary "." at the section headings of introduction, results, and discussion.
We have corrected this error in the revised version of the manuscript.
Round 2
Reviewer 2 Report
None.